# Transient Congenital Complete Heart Block: A Case Report

**DOI:** 10.3390/children8090790

**Published:** 2021-09-09

**Authors:** Ying-Tzu Ju, Yu-Jen Wei, Ming-Ling Hsieh, Jieh-Neng Wang, Jing-Ming Wu

**Affiliations:** Department of Pediatrics, National Cheng Kung University Hospital, College of Medicine, National Cheng Kung University, Tainan 70403, Taiwan; ints.ju@gmail.com (Y.-T.J.); garlicwei@gmail.com (Y.-J.W.); mactep_8@hotmail.com (M.-L.H.); jiehneng@mail.ncku.edu.tw (J.-N.W.)

**Keywords:** congenital, complete heart block, neonate, transient

## Abstract

Congenital complete heart block is defined as a complete atrioventricular block occurring prenatally, at birth, or within the first month of life. Congenital complete heart block has a high mortality rate, and in infants with normal heart morphology, it is often associated with maternal connective tissue disease. In these latter cases, neonatal congenital complete heart block is usually irreversible. We present a rare case of a female neonate who had bradycardia noted at a gestational age of 37 weeks. Her mother had no autoimmune disease history. She had no structural heart disease, and the serology surveys for autoantibodies including SSA/Ro and SSB/La were all negative. Without intervention or medication, her congenital complete heart block completely recovered to a normal sinus rhythm within 5 days. The cause of the transient congenital complete heart block was unknown in this case.

## 1. Introduction

Congenital heart block is defined as an atrioventricular block occurring prenatally, at birth, or within the first month of life, and it has a high mortality rate [1]. A congenital complete heart block is a complete blockade of impulses from the atrium to the ventricle. Thus, the ventricle beats independently in a junctional or ventricular rhythm, usually at approximately 40–80 beats per minute (bpm), and can result in a low cardiac output, fetal hydrops, and even death if left untreated [2]. The incidence of complete congenital heart block is approximately 1 in 22,000 live births. Approximately 50% of fetal cases with congenital heart block are associated with major cardiac structural defects, particularly left atrial isomerism, atrioventricular septal defects, or congenitally corrected transposition of the great arteries [3]. In infants with a normal heart structure, complete congenital heart block is strongly associated with maternal connective tissue disease, especially in infants born to a mother with positive SSA/Ro or SSB/La autoantibodies. Maternal autoantibodies and structural defects account for approximately 90% of complete congenital heart block cases, and the remaining 10% are regarded as idiopathic congenital heart blocks [4]. Neonatal complete atrioventricular block is usually irreversible, especially in infants whose mothers have SSA/Ro or SSB/La autoantibodies [4,5]. Approximately 63–89% of patients receive a permanent pacemaker before adulthood, and most are implanted within one month after birth [6,7]. Herein, we present a rare case of a normal neonate without underlying diseases; her complete congenital heart block completely recovered to a normal sinus rhythm within 5 days without invasive interventions or medication.

## 2. Case Presentation

A female neonate was born to a 29-year-old gravida 3 para 2 abortus 1 mother at the gestational age of 37 weeks with unremarkable prenatal laboratory results and an unremarkable second trimester anatomic ultrasound. She had no past history of autoimmune disease, and no medication was used during pregnancy. The fetal heartbeat (FHB) decreased to less than 60 bpm during a routine prenatal examination, and the infant was delivered via an emergency Cesarean section on the same day.

The baby cried well soon after birth, and the Apgar score was 8 at both 1 and 5 min after birth. The amniotic fluid was clear without meconium staining. The birth body weight was 2615 g, which was appropriate for the gestational age. She was admitted to the neonatal intensive care unit. On admission, bradycardia (heart rate: 70 bpm) was noted but the pulse was strong and no hypotension was detected (blood pressure: 70/29 mmHg). The peripheral saturation was 100% in room air. The electrocardiogram (ECG) showed a complete atrioventricular (AV) block with an atrial rate of 158 bpm and a ventricular rate of 75 bpm (Figure 1). Echocardiography showed a normal heart structure with a normal LV systolic function (EF: 59.7% by M mode measurement in parasternal long-axis view) and mild mitral regurgitation. The child had a normal hemogram and an unremarkable blood gas analysis. A cardiac biomarker survey showed a mildly elevated creatine kinase of 234 U/L and a creatine kinase isoenzyme (CK-MB) level of 8.65 ng/mL. High-sensitivity troponin T and N-terminal pro-brain natriuretic peptides were within the normal range. Immunological tests for serum SSA/Ro antibodies and SSB/La antibodies were both negative. The mother had no previous connective tissue disease history, and the anti-nuclear antibody (ANA) screening test was ANA-speckled and homogenous (1:160 positive) but C3, C4, and anti-dsDNA were within the normal ranges.

The patient was hemodynamically stable under continuous ECG monitoring in the neonatal intensive care unit. The heart rate was between 63 and 93 beats per minute without any signs of hypoperfusion. She received neither respiratory support nor inotropic agents, and there was no metabolic acidosis. A temporary pacemaker implantation, although initially planned, was eventually not performed because of the constant good clinical condition, sufficient perfusion, and acceptable heart rate. Repeated 12-lead ECG follow-up at 3 days of age indicated a change to a Mobitz type I second degree AV block with an atrial rate of 144 bpm and a ventricular rate of 60–90 bpm (Figure 2). To differentiate this rhythm, atropine was injected, and both the atrial and ventricular rates transiently increased. On the following day, the ECG revealed prolonged PR intervals without interruption of atrial-ventricular conduction, demonstrating passage to a first degree AV block. A normal sinus rhythm with a heart rate of 125 bpm was restored at 5 days of age (Figure 3). A follow-up, an outpatient visit at 1 month of age revealed a normal sinus rhythm and a heart rate of approximately 130–160 bpm.

## 3. Discussion

Complete congenital heart block increases the risk of morbidity and mortality in utero and in the first few months of life (15–30%) [2,8]. Complete congenital heart block can result in a low cardiac output, hydrops fetalis, and intrauterine fetal death (IUFD) [2]. In the literature, the rates of IUFD range from 9% to 45% and are highly associated with hydrops fetalis [8,9]. Treatments for a complete prenatal congenital heart block—including dexamethasone, intravenous immunoglobulin (IVIG), beta-adrenergic medication, and plasmapheresis—have been reported, but there is still no consensus on their efficacy [4]. Complete congenital heart block patients may have impaired cardiac function or low cardiac output. Early temporary pacing as bridging to a permanent pacemaker implantation in high-risk patients reduces the adverse consequences of profound bradycardia and even asystole [8]. Intravenous isoproterenol, atropine, epinephrine, and dopamine were also reported [1]. In the AHA 2012 guidelines, a permanent pacemaker was suggested for symptomatic complete congenital heart block infants, nonreversible AV nodal disease, and asymptomatic infants with a ventricular rate less than 55 bpm or less than 70 bpm in complete congenital heart block with major cardiac structural defects [1,10]. Pacemaker implantation is used in approximately 12–53% of neonates with complete congenital heart block, and most of them require long-term pacing [1,9,10,11]. Complications can occur after the implantation of pacemakers in up to 25% of cases including infections; cardiac perforations and thromboembolisms; lead fractures [12]; and, over the long-term, pacemaker-induced myocardial dysfunction. In our case, the patient was clinically asymptomatic without any low cardiac output signs after birth. We opted against temporary or permanent pacemaker insertion and continued to monitor the patient on the first day, although temporary pacing would have been considered in case a deterioration occurred. The clinical course spontaneously improved in the following 5 days, and the patient was discharged without medication or pacemaker implantation.

In SSA/Ro- or SSB/La-positive mothers, the risk of developing complete congenital heart block lies at about 1–2%. The mother may have symptoms of systemic lupus erythematosus or Sjögren’s syndrome or may be asymptomatic [8]. It is understood that maternal IgG autoantibodies cross the placenta and bind to fetal cardiac myocytes, causing inflammation and fibrosis of the conduction system [13]. In fetal and neonatal lupus erythematosus, heart block is usually diagnosed between 18 and 24 gestational weeks. The severity of the heart block depends on the scarring of the conduction system. Several specific maternal HLA antigen patterns may also be related to Sjögren’s syndrome or complete congenital heart block [14]. A complete AV block caused by maternal autoimmune autoantibodies is usually irreversible. There has only been one previous case report of a spontaneous and complete recovery in a mother with previously undiagnosed SLE. The cardiac rhythm reverted back to a normal sinus rhythm within 12 h of life, and both the mother and the infant had positive anti-SSA/Ro and SSB/La antibodies [15]. Our case and her mother had neither a history nor serology evidence of connective tissue disease, although the ANA titer was slightly high and the anti-SSA/Ro and anti-SSB/La data were unknown. Most autoimmune-related complete congenital heart blocks are irreversible and occur after the second trimester. Our case had a transient complete congenital heart block onset in the third trimester and spontaneously returned to normal sinus rhythm 5 days after birth so the etiology might have been different from a typical autoantibody-related complete congenital heart block. A possible explanation is that the mother might have had a subclinical autoimmune disease with autoantibodies causing a fetal AV node injury. After the delivery, the maternal autoantibodies disappeared and the AV node conduction returned to normal.

Reports of a spontaneous recovery from idiopathic complete congenital heart block are scarce. Four fetal cases of a transient heart block at 18 to 35 gestational weeks were reported in 2005. The heart blocks were diagnosed by fetal ultrasonography and were spontaneously resolved in 4 weeks. None of these mothers or infants had connective tissue disease, and the SSA/Ro and SSB/La antibodies were both negative. Maturation of the balance between the parasympathetic and the sympathetic nervous systems or myocardial ischemia might be the cause, but the cause of the transient fetal heart blocks was still unknown in this report [16]. The parasympathetic activity increases with fetus maturation during the third trimester and in newborns up to 3–6 months of age [17]. Mild hypoxia during the birth process, tactile stimulation, or cold temperatures can result in a vagal response and in lowering of the newborn’s heart rate. It was reported that one term-baby had sinus bradycardia at 52 h of life and spontaneously recovered 3 days later without intervention as the sympathetic system matured in the first few days of life [18]. In our case, an imbalance in the parasympathetic and sympathetic systems of the AV node could also explain the transient complete congenital heart block. Transient complete congenital heart block cases are rare, and examinations and the follow-up times of these case were limited. Therefore, the cause of the transient complete congenital heart block in our case remains unknown. Due to the possible spontaneous recovery of complete congenital heart block, a careful clinical decision-making is crucial in such cases to prevent unnecessary treatment in an asymptomatic patient without signs of low cardiac output.

## 4. Conclusions

In this case report, the infant had a complete congenital heart block in the third trimester and spontaneously reverted back to a normal sinus rhythm 5 days after birth. Although the serology surveys for autoantibodies were negative, undetected transplacental maternal autoantibodies or unreported events might have been possible causes of the transient complete congenital heart block. The presented case demonstrates that spontaneous recovery of complete congenital heart block is possible in infants without detectable autoantibodies.

## Figures and Tables

**Figure 1 children-08-00790-f001:**
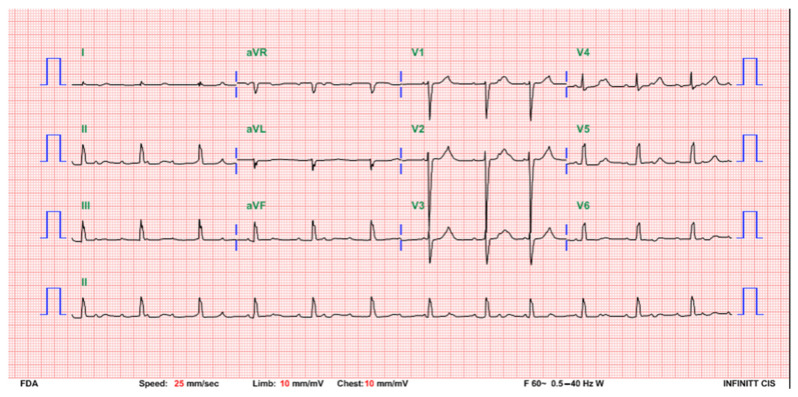
A postdelivery ECG indicating a complete AV block with an atrial rate of 158 beats per minute and a ventricular rate of 75 beats per minute.

**Figure 2 children-08-00790-f002:**
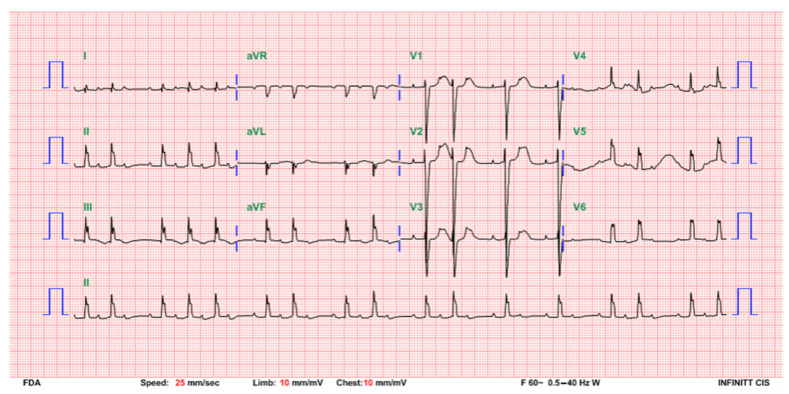
ECG at 3 days of age indicating a Mobitz type I second-degree AV block with an atrial rate of 144 bpm and a ventricular rate of 60–90 bpm.

**Figure 3 children-08-00790-f003:**
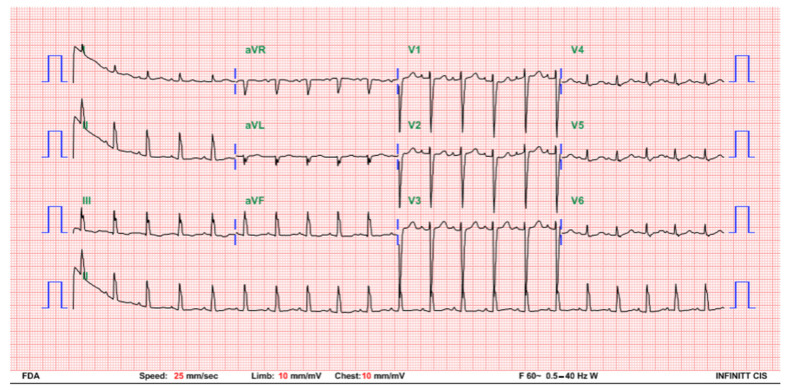
ECG at 5 days of age indicating a normal sinus rhythm with a heart rate of 125 bpm.

## Data Availability

All relevant data are within the manuscript.

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
