# Peer review of "Transient Congenital Complete Heart Block: A Case Report"

_children, 2021, doi:10.3390/children8090790_

Round 1

Reviewer 1 Report

This case report summerizes a rare case of transient congenital complete heart block. Complete heart block is rare and usually associated with strutural anomalies or maternal autoantibodies. This is definitely an unusual presentation of a rare phenotype. Review comments: 1) The manuscript needs english language editing. E.g. "Baby was discharged at 2 weeks old" There are different times used - please correct (e.g. "amniotic fluid is clear" and in the next senctence "body weight was..." 2) Please give more concise information about clinical presentation: pulse status on initial examination. And need for invasive/non invasive ventilation (respiratory support). Signs for hydrops? 3) please give more information about Echocardiography: RV and LV Function, regurgitations? Please comment on endocardial fibroelastosis 4) You have done testing for autoantibodies. However do not comment on HLA. Mothers of CHB children differ in some HLA class I alleles, and especially in HLA haplotypes, from mothers of healthy children. Mothers with HLA A1, Cw7, B8 and without B15 are at particularly high risk of having CHB children. Please see attached literature: Role of HLA in congenital heart block: susceptibility alleles in mothers M-K Sire et al. https://doi.org/10.1191/096120399678847399 5) Can you please comment on maternal medication, as this could cause specific symptoms. 6) Please comment on the outcome and the performed follow-up of the patient. 7) This is definitely an interesting case, however intrauterine asphyxia seems an unlikely for a child that has been otherwise clinically fine. Please critically review your conclusion or discuss this statement further in your discussion.

Author Response

Dear reviewer,

Our reply is as followings

  1. Had fixed that. Thank you (Line 40, 88)
  2. The patient had good activity, strong pulse and pink skin color after birth. No signs of hydrops fetalis. She did not need respiratory support and inotropic agent support. (Line 43-44)
  3. The patient had fair LV wall motion with fair LVEF 59.7% and mild mitral regurgitation. No cardiomegaly nor endocardial fibroelastosis were noted. (Line 66-67)
  4. The mother’s HLA was not checked. (Line 126)
  5. The mother had no medication history. Fetal bradycardia was noted at gestational age 37 weeks and she received emergent Cesarean section, so she did not receive medical therapy for fetal bradycardia. (Line 36)
  6. The patient had OPD visit follow up at 1 month old. Her EKG showed normal sinus rhythm and the growth was well. (Line 88-89)
  7. Asphyxia or subclinical maternal autoantibodies were possible explanation. But the cause of transient complete heart block of this case was still unknown. (Line 175-233)

Thank you for your comment!

Reviewer 2 Report

The Authors did not provide very important information  i.e. how was the SF and EF (Simpson) in the echo study? Whar was the CO results during the following days of observation etc. What was the Holter ECG results? It isn't known what the min, mean and max HR was, whether there were any pauses etc.  It was not stated whether temporary pacing was considered.  In addition, the figures (nice in fact) show only the instantaneous ECG image at THAT moment.

Author Response

Dear reviewer,

The patient’s left heart and right heart function were fair with LVEF 59.7% (M mode) (Line 67)

The cardiac function was good initially and the patient had no low cardiac output, no hypoperfusion signs nor metabolic acidosis, so we did not follow up the CO in the next few days.

The patient was monitoring in NICU with continuously EKG monitor, so we did not arrange a Holter EKG exam. The cardiac rhythm was the same as 12-lead EKG provided. The HR was 63 to 93 /min in first day. The increased HR may be artifact because feeding or changing patient position may disturbed the monitoring sensor. No pause or ectopic rhythm were recorded. (Line 81)

The temporary pacemaker was not considered initially for the patient was asymptomatic and the cause of complete AVB was uncertain. So we observation first. In the next day morning, the rhythm changed so we kept waiting and the rhythm returned to sinus rhythm spontaneously. There was no low cardiac output signs and no metabolic acidosis during these observation days. (Line 82, 161-165)

Thank you for your comment

Round 2

Reviewer 1 Report

This case report describes a rare case of a transient complete heart block in a neonate without maternal autoantibodies.

The authors made an effort to implement the recommended changes. and comments from the first review.

Comments:

  • The english language has not improved in the revised manuscript. There are several grammatical and syntax errors.
  • The introduction is covers now the same information as the abstract and does not give the reader any more background information.
  • Abbreviations are not properly described/ introduced, e.g. NICU or introduced and then not consequently followed e.g. CCHB
  • Line 69 "pacemaker was not indicated", no explanation given about decision making and guidelines that were applied.
  • A lot of information given in the discussen could be transferred to the introduction.
  • In the discussion; the authors repeat a lot of the case presentation. They cite literature without critically discussing the implication for their patient.

Author Response

We had sent the article for English editing. 
We had transferred some information to introduction. 
The pacemaker was not indicated initially for stable clinical condition and relative acceptable heart rate. Although we were ready for temporary pacemaker implantation if deterioration. (Line 139-140)

Thank you for your comment!!!

Reviewer 2 Report

The authors take note my comments. Now, the article is valuable

Author Response

Thank you for your comment!!

Round 3

Reviewer 1 Report

Third review:

The manuscript has improved from the previous version.

A few more comments:

  1. Please introduce CCHB after first mentioned in the introduction and continue to use abbreviation e.g. line 31 etc.
  2. Line 75: “Patient was stable and had an ECG monitor in the neonatal unit” – please rephrase: The patient was hemodynamically stable under continuous ECG monitoring in the neonatal intensive care unit
  3. Line 79 cross of "relatively"
  4. Line 80 Patient was under continuous ECG monitoring and then you speak about serial ECG. You probably mean 12-lead ECG – please specify. E.g. Repeated 12-lead ECGs showed….
  5. Line 84: indicated a change to first degree block. Is it a first degree block or not? Please be more specific.
  6. Discussion please use your abbreviation CCHB again
  7. Line 122 – we opted against temporary or permanent pacemaker insertion and continued to monitor …. (instead of we maintained an observation)
  8. Line 152 : returned to normal sinus rhythm five days after birth
  9. Line 211 remains unknown
  10. You could add one comment at the end: that careful clinical decision-making is crucial in this case to prevent unnecessary treatment in a patient with no symptoms for low cardiac output…. I think that would be a nice summary.
  11. Line 214 spontaneously converted back to normal sinus rhythm 5 days after birth
  12. Line 171 and in the conclusion: I still cant support the argument with the asphyxia of the av node. In a structural normal heart with normal coronary anatomy (on echo). No other signs of asphyxia I think this is very unlikely. And I would definitely suggest removing this from the conclusion of the manuscript. And there is no literature given to support your hypothesis.

Author Response

Dear Editors and Reviewers:

Thank you for your detailed review and suggestion for our work. Please see each point to point response to reviewer’s comments as follows:

Q1: Please introduce CCHB after first mentioned in the introduction and continue to use abbreviation e.g. line 31 etc.

Response: Thanks for your suggestion. We use the abbreviation to replace “congenital complete heart block” in the text .We keep some “congenital heart block” in the text due to these sentences describes all forms of heart block.

Q2: Line 75: “Patient was stable and had an ECG monitor in the neonatal unit” – please rephrase: The patient was hemodynamically stable under continuous ECG monitoring in the neonatal intensive care unit

Response: Thanks for your suggestion. We changed the sentence as your recommendation in Line 70.

Q3: Line 79 cross of "relatively"

Response: We corrected the sentence in Line 74.

Q4: Line 80 Patient was under continuous ECG monitoring and then you speak about serial ECG. You probably mean 12-lead ECG – please specify. E.g. Repeated 12-lead ECGs showed….

Response: Thanks for your detailed reading. We adjusted the sentence in Line 75

Q5: Line 84: indicated a change to first degree block. Is it a first degree block or not? Please be more specific.

Response: Thanks for your comments. The follow-up ECG revealed prolonged PR interval without interruption. We added this information in Line 78.

Q6: Discussion please use your abbreviation CCHB again

Response: We replaced the “congenital complete hear block” in the discussion section to “CCHB” in line 98, 99, 164.

Q7: Line 122 – we opted against temporary or permanent pacemaker insertion and continued to monitor …. (instead of we maintained an observation)

Response: Thanks for your suggestion. We modified the sentence on Line 121—122.

Q8: Line 152 : returned to normal sinus rhythm five days after birth

Response: Thanks for your suggestion. We modified the sentence on Line 143.

Q9: Line 211 remains unknown

Response: Thanks for your suggestion. We modified the sentence on Line 164.

Q10: You could add one comment at the end: that careful clinical decision-making is crucial in this case to prevent unnecessary treatment in a patient with no symptoms for low cardiac output…. I think that would be a nice summary.

Response: Thanks for your suggestion. We added a summary based on your suggestion in the end of this section.

Q11: Line 214 spontaneously converted back to normal sinus rhythm 5 days after birth

Response: Thanks for your suggestion. We modified the sentence on Line 167.

Q12” Line 171 and in the conclusion: I still cant support the argument with the asphyxia of the av node. In a structural normal heart with normal coronary anatomy (on echo). No other signs of asphyxia I think this is very unlikely. And I would definitely suggest removing this from the conclusion of the manuscript. And there is no literature given to support your hypothesis.

Response: Thanks for your suggestion. We modified the sentence in the Discussion and conclusion section. To highlight the experience we learned, we reemphasized the summary as Q10 suggested in the end of conclusion.